# VNT4, a Derived Formulation of Glutacetine® Biostimulant, Improved Yield and N-Related Traits of Bread Wheat When Mixed with Urea-Ammonium-Nitrate Solution

Victor Maignan [1,2,*], Raphaël Coquerel [1], Patrick Géliot [2] and Jean-Christophe Avice [1,*]

[1] UFR des Sciences–Département Biologie et Sciences Terre, UNICAEN, INRAE, UMR EVA, Normandie Univ., SFR Normandie Végétal FED4277, Esplanade de la Paix, F-14032 Caen, France; raphael.coquerel@unicaen.fr
[2] Via Végétale, F-44430 Le Loroux-Bottereau, France; pgeliot@viavegetale.com
* Correspondence: vmaignan@viavegetale.com (V.M.); jean-christophe.avice@unicaen.fr (J.-C.A.)

**Abstract:** Optimizing nitrogen use efficiency (NUE) could mitigate the adverse effects of nitrogen (N) fertilizers by limiting their environmental risks and raising agronomic performance. We studied the effects of VNT4, a derived formulation of Glutacetine® biostimulant, mixed with urea-ammonium-nitrate solution (UAN) on the growth, N-related traits and agronomic performance of winter wheat (*Triticum aestivum* L.). The experiment was performed under six contrasting field conditions over two years in Normandy (France), including a site where $^{15}$N labelling was undertaken. Taking into account all the sites, we report that VNT4 significantly improved grain yield (+359 kg ha$^{-1}$), total grain N and NUE. VNT4 application improved growth during tillering and stem elongation (+10.7%), and N and $^{15}$N uptake between tillering and maturity (+7.3% N and +16.9% $^{15}$N) leading to a higher N accumulation at maturity (+9.3% N). This N mainly originated from fertilizer (+19.4% $^{15}$N) and was assimilated after the flag leaf stage in particular (+47.6% $^{15}$N). These effects could be related to maintenance of physiological functions of flag leaves as suggested by the enhancement of their nutrient status (especially S, Zn and Mo). The adoption of VNT4 as a UAN additive is an efficient agronomic practice to enhance wheat productivity under an oceanic temperate climate.

**Keywords:** *Triticum aestivum*; nitrogen use efficiency; nitrogen fertilizer; UAN; VNT4; $^{15}$N labelling

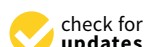



## 1. Introduction

Nitrogen (N) fertilization is a major input in intensive agricultural systems. It has been estimated that 48% of the global population was fed via N fertilizers in 2008 [1]. In addition, the use of N fertilizers has increased eight-fold since 1961 due to the growing demand for agricultural products and especially crops [2]. Thus, N fertilizer management is one of the most challenging tasks for cereal farmers with agronomic, economic and environmental complexities [3]. Indeed, N chemical fertilizers are essential to ensure yield and quality of the harvested products [4], but N fertilizers are involved in many environmental issues. Nitrate (NO$_3^-$) leaching and gaseous emissions of ammonia (NH$_3$) and nitrous oxide (N$_2$O) are harmful N pollutants, which also contribute towards lowering N use efficiency (NUE) when lost from agroecosystems [5,6]. Nitrate leaching is an important contaminant of groundwater and surface water resources [7]. Ammonia contributes to soil acidification, eutrophication/dystrophication and the formation of particles of a mean aerodynamic diameter smaller than 2.5 μg (PM2.5) that have been related to human respiratory distress [8]. Nitrous oxide is a greenhouse gas (GHG) with a global warming potential 298 times higher than CO$_2$ in the atmosphere, and it also contributes to the destruction of stratospheric ozone [9]. Thus, to mitigate these environmental issues, sustainable agricultural practices need to be implemented.

Strong efforts have been made to improve NUE through breeding programs [10–12] and the enhancement of agronomic practices. For example, it is well documented that cover

crops reduce nitrate leaching [13]. Besides, N interacts with other elements, which has strong repercussions on NUE. Among them, sulfur (S), molybdenum (Mo) and zinc (Zn) have synergistic effects with N [14–16]; it is therefore important to have a good nutritional balance in wheat (*Triticum aestivum* L.) in order to ensure high NUE. During the crop cycle, it is recommended to divide fertilizer additions into four applications [17], to spray N fertilizer at anthesis to stimulate N metabolism [18,19] or to apply late N nutrition [20] in order to avoid $NH_3$ volatilization and GHG emissions while enhancing NUE. These recommendations highlight that is necessary to apply the right dose at the right application time to increase the congruence between N supply and crop N demand and thus better manage N fertilization. Indeed, NUE, which is expressed as the harvestable yield per N supply [21], is correlated with the capacity of plants to take up, transport [22], store and remobilize N [23,24].

Another lever to improve NUE is based on enhanced-efficiency fertilizers (EEFs). These efficient fertilizers frequently contain inhibitors of urease and/or nitrification [25]. Urease inhibitors are compounds that delay the hydrolysis of urea, whereas nitrification inhibitors are compounds that delay bacterial oxidation of ammonium by depressing the activities of nitrifiers in the soil [26]. N-(n-butyl) thiophosphoric triamide (NBPT), which is a structural analogue of urea, is one of the most studied urease inhibitors [27]. Regarding nitrification inhibitors, 3,4 dimethylpyrazole phosphate (DMPP) is widely used in Europe [28]. Both kinds of inhibitors have been shown to reduce $NH_3$ and $N_2O$ loss while improving yield and NUE [25,29]. Nevertheless, these chemical inhibitors also affect soil microorganism activity and plant metabolism [30–32]. This is why environmentally friendly alternatives such as controlled-release fertilizers (CRFs) by coating fertilizer [33] and/or biostimulants to form EEFs have been developed [18]. CRFs allow the slow delivery of nutrients, which can reduce the number of fertilizer applications and the total amount of fertilizer applied, resulting in increased fertilizer use efficiency. For N, CRFs such as S- or resin-coated urea can reduce N losses to the environment and improve grain yield in rice [33,34]. A recent meta-analysis indicated that CRF application significantly increased soil organic carbon, total N and available N by 5.93%, 3.89% and 13.98%, respectively, which increased the yield and NUE in wheat, maize and rice [35]. Due to their ability to slowly release nutrients, it was recently demonstrated that biochar-based controlled treatment also improved maize (*Zea mays* L.) grain yield when applied alongside inorganic N on sandy soils with poor physical and chemical properties [36]. Other EEFs based on ecological raw materials have been recently developed and include biostimulants such as humic acids, seaweed or fungal extracts [37,38].

Indeed, according to the EU regulation 2019/1009, "a plant biostimulant is a product that stimulates plant nutrition processes independently of the product's nutrient content with the sole aim of improving one or more of the following characteristics of the plant or the plant rhizosphere: (a) nutrient use efficiency; (b) tolerance to abiotic stress; (c) quality traits; (d) availability of confined nutrients in the soil or rhizosphere". Biostimulants influence phenotypic traits and improve yield mainly by enhancing nutrient uptake and assimilation [39,40]. This is why bioactive substances and microorganisms are of interest for improving nutrient use efficiency [41,42]. Regarding N nutrition, it is well documented that biostimulants can improve N acquisition [43,44] and plant N metabolism [45], for example, by delaying senescence [46] or increasing remobilization efficiency [18]. In addition, biostimulants are also modulated by N fertilizers [47,48]. This is the capacity of biostimulants to improve NUE, which is the main reason that they have been gaining ground in agricultural systems [49]. These products can therefore be used to complement fertilizers [50]. Although the costs of these EEFs are usually higher than conventional fertilizers, their application can reduce the amount of residual nutrients in the soil and improve both crop yield and economic efficiency in crops such as wheat.

Wheat is one of the most important cereal crops and needs high N fertilizer inputs in order to achieve high yield and protein content in the grain [51,52]. The dynamics of nutrient exchange have been well studied in wheat [53], especially related to



N [23,54–56]. Despite strong efforts to breed new cultivars and to improve agronomic practices that enhance NUE, it has been estimated that only one third of applied N is recovered in the harvested component of grain crops [57]. This means that it is necessary to find new methods adapted to different cropping systems. Consequently, both local and crop-specific solutions will be required for NUE improvement. In this context, EEFs could be an interesting way to increase fertilizer use efficiency while avoiding N loss and improving NUE in wheat. Many studies have reported that biostimulants could ameliorate wheat productivity and quality. For example, a fulvic acid extract enhanced soil properties and induced better growth, yield and nutritional status in wheat [58]. *Bacillus* strains have been shown to increase bread wheat performance under different N supply [48], while marine and fungal biostimulants led to a small increase in grain yield in durum wheat [44]. Mixed with N fertilizer, biostimulants, such as Glutacetine®, have improved yield and N-related traits in winter wheat under controlled conditions [18] and under contrasting field trials [59]. Furthermore, a ureic-based controlled-release fertilizer formulated with water-soluble polymeric coatings enriched with humic acids or seaweed extracts also enhanced yield in wheat [37]. Nevertheless, there has been a lack of research into combining biostimulants and N fertilizers to improve wheat productivity and NUE in temperate-climate regions. Moreover, the performance of these products is determined by the dose and the date of application, and there is a substantial influence from the soil and weather conditions [60]. Designing and developing new EEFs is thus an essential process that requires accurate testing of the products' effects on yield and N-related traits under field conditions.

The goal of this study was to examine the effects of VNT4, a derived formulation of Glutacetine®, mixed with urea ammonium nitrate solution (UAN, forming an EEF) on grain yield and N-related traits in winter wheat under field conditions in a temperate climate. Experiments were carried out at six different sites in Normandy (France) for two years to evaluate the possible beneficial effects of this EEF applied on different cultivars under optimal or suboptimal N conditions and to understand its influence on NUE and yield components. Labelled $^{15}NH_4^{15}NO_3$-CO $(^{15}NH_2)_2$ was applied in parallel microplots to study the main processes involved in N uptake, assimilation and remobilization, with the aim of discovering the most efficient strategies.

## 2. Materials and Methods

### 2.1. Field Experiment Design under Contrasting Conditions (Experiment 1)

In 2018 and 2019, experiments were carried out at six sites, each with a different cultivar of winter wheat (*Triticum aestivum* L., cv. Sacremento, Libravo, Chevignon in 2018; Adoration, Boregar and Extase in 2019) in Normandy, France (see Table S1 for more detail), which is classified as having an oceanic temperate climate. Plot size was 24 m$^2$ (12 m × 2 m) and three and four replicates (plots) were used per treatment at each site during the first and the second year, respectively. The experimental design was a randomized complete block design with 42 and 48 microplots in 2019 and 2020, respectively, and data presented in this work is a subset of treatments. During the growing period, systematic applications of plant protection treatments were carried out. Weather conditions during crop cycle were from 11.7 to 12.4 °C for average temperature and from 585 to 883 mm for total rainfall between September 2018 and August 2020 (Figure S1). Previous crops and soil properties were very different at each site (Table S1).

At each site, all plots were fertilized with 30 kg S ha$^{-1}$ at the tillering stage during the first year (BBCH 29) [61]. N fertilizer applications were intended to replicate farmer practices and were calculated using the so called "Méthode du Bilan" described by the French COMIFER committee [62]. This method measures the amount of mineral N in the soil in February to estimate total soil N supply. With the estimation of total plant needs, the method allows application of an N dose that is close to the crop demand (Table S2). Because the N content given by the biostimulant was very low, it was neglected for the calculation of N fertilization. Each plot was fertilized with 39% UAN. N fertilizer was applied at 152, 210 and 152 kg N ha$^{-1}$ at sites 1, 2 and 3 in 2019, respectively. In 2020, we

reduced the dose by 40 kg N ha$^{-1}$ from the optimal amount calculated to create suboptimal N conditions: 175, 92 and 179 kg N ha$^{-1}$ were applied at sites 4, 5 and 6, respectively (Table S2). The first fertilizer application was undertaken at tillering (BBCH 20-25), the second at stem elongation (BBCH 31), and the last at the flag leaf stage (BBCH 39, Table S2). For the last N application, we followed the new N requirement indicators ("bq" coefficient) indicating the amount of N to apply after stem elongation for growing each bread wheat cultivar with the dual objective of optimum yield and grain protein content in line with market requirements [17]. For the biostimulant treatment, 5 L ha$^{-1}$ of VNT4 formulation (see Table 1 for more details) were mixed with UAN solution before each N application, leading to a global application of 15 L ha$^{-1}$. In parallel, NBPT was mixed at a concentration of 0.2 % in UAN at each N application (2 mL NBPT for 1 L of UAN) to compare VNT4 with this urease inhibitor. Each number was counted in each plot of the six sites and grain and straw were collected using a combine harvester. All samples were dried (60 °C) for 48h and then ground to a fine powder using a Retsch MM200 mixer mill (Retsch, Eragny sur Oise, France) for N analysis.

**Table 1.** Composition of the VNT4 formulation.

| Component | | Concentration |
|---|---|---|
| Glutamic acid (%) | | 3.6 |
| Organic acids (%) | | 7.4 |
| Total soluble sugars (g L$^{-1}$) | | 34.4 |
| Elements (%) | Cl | 18.2 |
| | Ca | 13.4 |
| | C | 11.5 |
| | N | 0.88 |
| | K | 0.29 |
| | Na | 0.18 |
| | Mo | 0.18 |
| Elements (ppm) | S | 140 |
| | B | 20 |
| | Mg | 11 |
| | Si | 11 |
| | P | 10.2 |
| | Cu | 1.1 |
| | Ni | 0.6 |
| | Co | 0.4 |
| | Zn | 3.5 |
| | Se | 0.06 |

### 2.2. $^{15}$N Experiment under Field Conditions (Experiment 2, Site 4)

In 2019, winter wheat (*Triticum aestivum* L., cv. Adoration) was sown on 28 December in the north of Normandy, France (49°25′ N, 0°61′ W, Site 4, Table S1). Plot size was 24 m$^2$ (12 m × 2 m) and a microplot (1.5 m x 1 m) inside this plot was treated with $^{15}$N solution. Four replicates (plots) were used per treatment, and the plots were randomized in four main blocks. N fertilization was the same as in Experiment 1: 175 kg N ha$^{-1}$ of UAN were split into three applications (Table S2). Each microplot received was labelled UAN ($^{15}$NH$_4$$^{15}$NO$_3$-CO ($^{15}$NH$_2$)$_2$, 98 atom% $^{15}$N (Cambridge Isotope Laboratories, Inc., Tewksbury, USA) with 2.5 $^{15}$N atom% excess applied with or without VNT4 (5 L ha$^{-1}$ with each N fertilization) at BBCH 2, 31 and 39. The first application at tillering was split between BBCH 20 and 25 due to the low biomass. A solution of 2.5 L of each treatment was prepared and homogenously applied over the soil surface of the microplots. Additionally, a control treatment with no fertilizer was included. These microplots were located at the end of the 24 m$^2$ plots. An area of 0.25 m$^2$ was harvested after each N fertilizer application (just before the next fertilization) and 0.25 m$^2$ at maturity to follow the growth throughout the

crop cycle and calculate the Harvest Index (HI, HI = grain biomass/aerial biomass). At the heading, flowering, seed development and maturity stages, 20 flag leaves were separated in the 24 m$^2$ plots and at maturity stage, flag leaves, grain and straw were harvested for detailed analyses. All samples were then dried in an oven (60 °C) for 48 h and ground to a fine powder using a Retsch MM200 mixer mill (Retsch, Eragny sur Oise, France) for elemental analysis.

### 2.3. N and $^{15}$N Analyses

For N and $^{15}$N analysis in shoots, flag leaves, grain and straw, an aliquot of each dried sample was weighed and placed into tin capsules using a microbalance. The total N concentration was determined with a continuous flow isotope ratio mass spectrometer (IRMS, Horizon, NU Instruments, Wrexham, United Kingdom) linked to a C/N/S analyser (EA3000, Euro Vector, Milan, Italy). For Experiment 1, total grain N was then calculated (kg N ha$^{-1}$) for all sites. For Experiment 2, N and $^{15}$N uptake between each fertilizer supply, the total N and $^{15}$N in aerial parts at each harvest, and the N harvest index (NHI, NHI = total grain N/total N in aerial parts) and $^{15}$N Harvest Index ($^{15}$NHI, $^{15}$NHI = total grain $^{15}$N/total $^{15}$N in aerial parts) were calculated. When calculation of $^{15}$N uptake resulted in a negative value, it was considered to represent 0 mg N m$^{-2}$.

### 2.4. Elemental Analysis in Flag Leaves

The whole ionome in wheat flag leaves was quantified by inductively high-resolution coupled plasma mass spectrometry (HR ICP-MS, Thermo Scientific, WA, USA, Element 2$^{TM}$) with prior microwave acid sample digestion (Multiwave ECO, Anton Paar, les Ulis, France) (800 µL of concentrated HNO$_3$, 200 µL of H$_2$O$_2$ and 1 mL of Milli-Q water for 40 mg DW). For the determination by HR ICP-MS, all the samples were spiked with two internal-standard solutions of gallium and rhodium for final concentrations of 10 and 2 µg L$^{-1}$, respectively, diluted to 50 mL with Milli-Q water to obtain solutions containing 2.0% (*v/v*) nitric acid, then filtered at 0.45 µm using a teflon filtration system (Filtermate, Courtage Analyses Services, Mont-Saint-Aignan, France). Quantification of each element was performed using external standard calibration curves. The amount of a given element, E, was calculated from DW and element concentration (%E as % of DW) as: E = (%E × DW)/100.

### 2.5. Grain Yield Components and N Use Efficiency

After the harvesting of winter wheat, 1000 grain weight (TGW, g), grain number per spike, grain per square metre, specific weight (grain density, kg hL$^{-1}$) and moisture (%) were determined. Grain yields were then calculated with a standard moisture (15%) and NUE according to Moll et al. (1982) using the following equation: NUE (kg kg$^{-1}$) = Grain yield (kg ha$^{-1}$)/N supply (kg ha$^{-1}$) [21].

### 2.6. Statistical Analysis

All statistical analyses were performed with R Statistical Software version 3.3. The normality of data and the homogeneity of variance were verified using the Shapiro and Levene tests, respectively. When data followed a normal distribution, one-, or two-way analysis of variance (ANOVA) was used to evaluate the effect of site (S), VNT4 (treatment, T) and their interaction (S × T). The resulting variations in data are expressed as the mean ± standard error (SE) for n = 3 in 2018–2019 and n = 4 in 2019–2020. Significant differences of means between treatments were separated with Fisher's test ($p \leq 0.05$). Some sets of data were transformed using the Box–Cox method in order to obtain a normal distribution and apply ANOVA and Fisher's test. When variables did not satisfy normality due to culture effects, the non-parametric Kruskal–Wallis test was used ($p \leq 0.05$).

## 3. Results

*3.1. Experiment 1: Study of the Impact of VNT4 on Grain Yield, Yield Components and N-Related Traits under Six Contrasting Field Trials*

In the present work, we evaluated the effect of $3 \times 5$ L ha$^{-1}$ of VNT4, a derived formulation of Glutacetine® (Table 1), mixed with UAN, at the tillering, stem elongation and flag leaf stages, on the yield and N-related traits of winter wheat. The product was tested in six field trials for two years to access the biostimulant properties under contrasting conditions. We wanted to investigate whether application of VNT4 increased the efficiency of UAN, which is one of the most widely used N fertilizers.

### 3.1.1. Impacts of Year, Site and VNT4 on Grain Yield and Its Components

To better understand the impact of site, VNT4 and their interaction, a global ANOVA was first undertaken for grain yield and several yield components (spike number per square metre, grain number per spike, grain number per square metre, TGW and specific weight). Depending on the field conditions, grain yields ranged from 6.11 to 12.52 t ha$^{-1}$. As wheat was under optimal conditions in 2019 and under suboptimal conditions in 2020 (40 kg N ha$^{-1}$ less than the recommendation), significant variations in yields between years were expected. The results over two years including six sites indicated strong site effects for grain yield and all yield components ($p < 0.001$, Table 2). Indeed, yields were much lower in 2020 than in 2019 (Figure 1A) due to both reduction in spike number per square metre (Figure 1B) and grain number per spike (Figure 1C), leading to a decrease in the grain number per square metre in 2020 (Figure 1D). However, there was no consistent change in TGW between 2019 and 2020 (49.8 g and 50.5 g on average, respectively), whereas there were contrasting site responses each year (Figure 1E). Specific weights were more homogenous in 2020 (from 74.5 to 76.1 kg hL$^{-1}$) than in 2019 and ranged from 71.6 (at site 2) to 86.4 kg hL$^{-1}$ (at site 3, Figure 1F). Regarding site effects, we observed higher yields at sites 1 and 2 than at site 3 in 2019, while the lowest results in 2020 were at site 5 (Figure 1A). This is mainly explained by the reduction in spike number per square metre at site 3 in 2019 (Figure 1B). In 2020, the yield reduction at site 5 was not due to the number of grains per square metre (Figure 1D) but to the TGW, which was significantly lower compared to the other sites (Figure 1E).

**Table 2.** Effects of site (S) and/or biostimulant treatment (T: VNT4) on grain yield, yield components and N-related traits determined by two-way ANOVA (global ANOVA and for each year).

| Parameter | Global ANOVA | | | ANOVA 2019 | | | ANOVA 2020 | | |
|---|---|---|---|---|---|---|---|---|---|
| | Site | Treat. | S × T | Site | Treat. | S × T | Site | Treat. | S × T |
| Yield | *** | * | 0.951 | *** | *0.073* | 0.813 | *** | 0.238 | 0.875 |
| Spike m$^{-2}$ | *** | 0.289 | *0.050* | ** | 0.894 | 0.189 | *** | 0.261 | *0.074* |
| Grain spike$^{-1}$ | *** | 0.952 | 0.142 | * | 0.733 | 0.370 | ** | 0.857 | 0.123 |
| Grain m$^{-2}$ | *** | *0.065* | 0.994 | ** | *0.085* | 0.906 | 0.262 | 0.397 | 0.999 |
| TGW | *** | 0.267 | 0.455 | ** | 0.977 | 0.519 | *** | 0.117 | 0.318 |
| Specific weight | *** | 0.687 | 0.524 | *** | 0.636 | 0.433 | ** | 0.959 | 0.597 |
| %N in grain | *** | 0.148 | 0.672 | *** | 0.678 | 0.816 | 0.226 | 0.123 | 0.454 |
| %N in straw | *** | 0.291 | 0.995 | *** | 0.210 | 0.567 | * | 0.448 | 0.906 |
| Total grain N | *** | * | 0.835 | *** | 0.232 | 0.727 | ** | *0.089* | 0.575 |
| NUE | *** | * | 0.893 | *** | *0.075* | 0.819 | *** | 0.291 | 0.965 |

*, **, *** = significant at 0.05, 0.01 and 0.001, respectively, *p* > F for n = 3 in 2019 and n = 4 in 2020. TGW: Thousand Grain Weight; NUE: Nitrogen Use Efficiency.

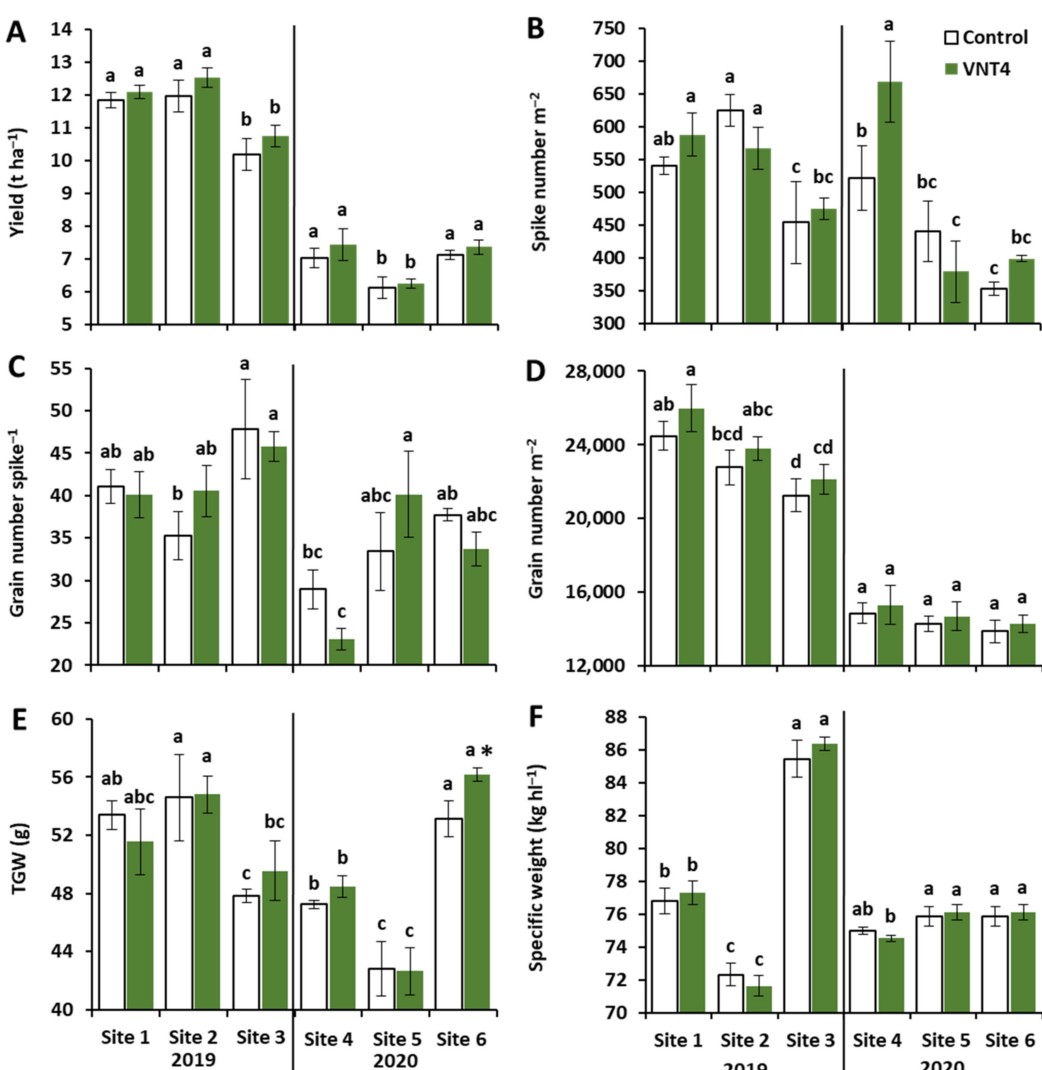

**Figure 1.** Influence of VNT4 on yield and its components in winter wheat under different site and year conditions. (**A**) Yield, (**B**) spike number per square metre, (**C**) grain number per spike, (**D**) grain number per square metre, (**E**) thousand grain weight (TGW) and (**F**) specific weight. Plant culture was carried out under six different field conditions in France, Normandy. N was provided at the tillering, stem elongation and flag leaf stages. 5 L ha$^{-1}$ of VNT4 formulation (see Table 1 for more details) were mixed with UAN before each N application, leading to a global application of 15 L ha$^{-1}$. Bars indicate means ± SE. Different letters denote significant differences in 2019 or 2020 according to Fisher's test ($p < 0.05$; n = 3 in 2019, n = 4 in 2020) and star denote significant differences between control and VNT4 according to a pairwise comparison (Fisher's test, *; $p < 0.05$, n = 4).

Interestingly, global ANOVA revealed a significant effect of VNT4 on yield ($p = 0.036$, Table 2), which increased by 463 kg ha$^{-1}$ in 2019 (+4.1%) and by 255 kg ha$^{-1}$ in 2020 (+3.8%, Figure 1A). Thus, globally, VNT4 improved yield by 359 kg ha$^{-1}$ (+4%). In parallel, NBPT also significantly increased yield compared to the control ($p < 0.05$, +308 kg ha$^{-1}$, +3.4%, Figure S2A). However, although VNT4 led to higher grain yields (+51 kg ha$^{-1}$) than NBPT, there was no significant difference between these two treatments (Figure S2A). In 2019, VNT4 had a slight effect on yield ($p = 0.073$, Table 2) with an amelioration of 246, 569 and 574 kg ha$^{-1}$ at sites 1, 2 and 3, respectively (Figure 1A). This improvement was either due to the increase in spike number per square metre (Figure 1B) or to a better grain number per spike (Figure 1C), leading to a better grain number per square metre irrespective of the site (Figure 2D and Table 2, $p = 0.065$). Indeed, there was an interaction

between site and treatment (S × T) for spike number per square metre ($p = 0.05$, Table 2). This component was increased by VNT4 at sites 1, 3, 4 and 6, whereas it was decreased at sites 2 and 5 (Figure 1B). Moreover, there was a slight S × T interaction the second year ($p = 0.074$, Table 2) mainly because VNT4 significantly improved spike number per square metre at site 4 (Figure 1B). In contrast to spike number per square metre, grain number per spike was improved at sites 2 and 5 (Figure 1C). Thus, VNT4 improved grain number per square metre at sites 1, 3, 4 and 6 by increasing the spike number per square metre, while at sites 2 and 5 grain number per spike improved (especially in 2019, $p = 0.085$, Table 2). Furthermore, there was no consistent change in TGW or specific weight after VNT4 application according to the global ANOVA (Table 2). However, VNT4 slightly enhanced TGW at site 6 (+5.7%, $p < 0.05$, pairwise comparison, Figure 1E). Altogether, these results indicated that VNT4 increased yield via different mechanisms depending on the site, but irrespective of the contrasting conditions (therefore, under both optimal and suboptimal N conditions). VNT4 enhanced spike number per square metre (especially at site 4), grain number per spike and TGW (especially at site 6).

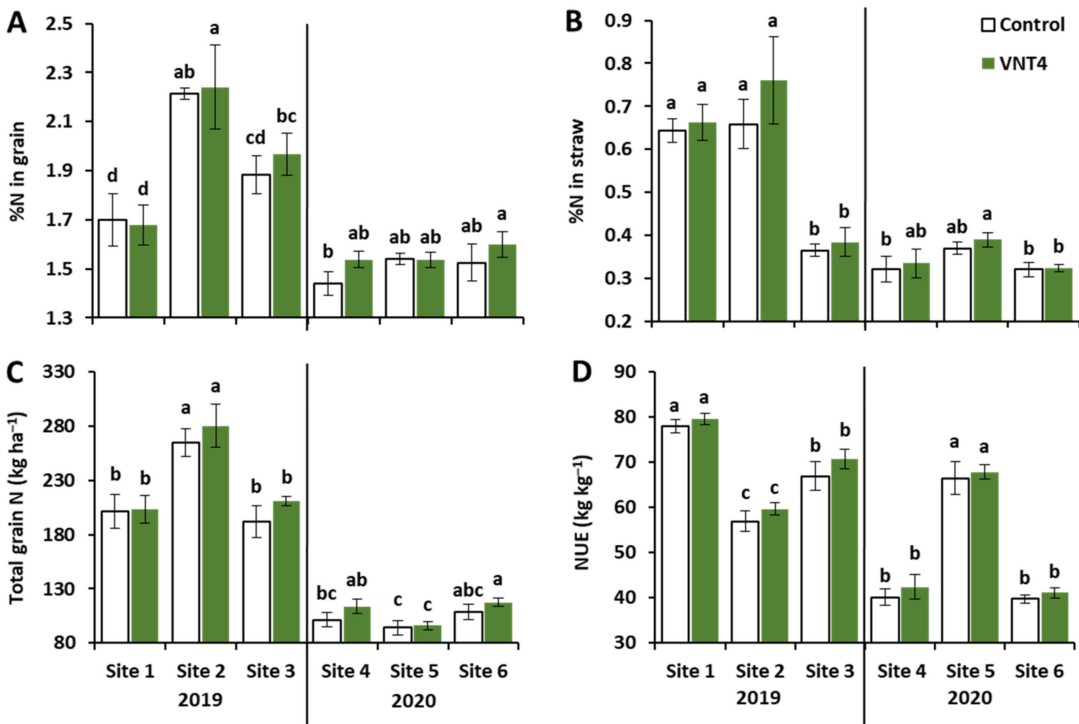

**Figure 2.** Influence of VNT4 on N-related traits in winter wheat under different site and year conditions. (**A**) N content in grain, (**B**) N content in straw, (**C**) total grain N and (**D**) NUE. Plant culture was carried out under six different field conditions in France, Normandy. N was provided at the tillering, stem elongation and flag leaf stages. 5 L ha$^{-1}$ of VNT4 formulation (see Table 1 for more details) were mixed with UAN before each N application, leading to a global application of 15 L ha$^{-1}$. Bars indicate means ± SE. Different letters denote significant differences in 2019 or 2020 according to Fisher's test ($p < 0.05$; n = 3 in 2019, n = 4 in 2020).

### 3.1.2. Effects of Year, Site and VNT4 on N-Related Traits

After the final harvest, global ANOVA revealed site effects ($p < 0.001$, Table 2) for the N concentration in grain and straw (Figure 2A,B). Regarding N content in grain, we observed lower concentrations in 2020 than in 2019 (Figure 1A), which was consistent with the N conditions for each year. Field conditions in 2019 resulted in strong differences between each site, with high N content in the grain at site 2, intermediate values at site 3 and a low N concentration at site 1 (Figure 2A), while there was no consistent change in 2020 (Table 2). However, there was a site effect in both years for N content in straw (Table 2). Indeed, the %N in straw was higher at sites 1 and 2 than site 3 in 2019, while

it was better at site 5 than sites 4 and 6 in 2020 ($p < 0.01$, Figure 2B). The year and site effects were also consistent for total grain N and NUE (Figure 2 and Table 2). Because conditions were optimal in 2019 and suboptimal in 2020, the total grain N was greatly reduced in 2020 compared to 2019 (Figure 2C). Site conditions also had an important role in the accumulation of N in the grain. In 2019, total grain N was significantly higher at site 2 than sites 1 and 3, which seemed to be correlated with N content in the grain (Figure 2A,C). In 2020, total grain N was significantly lower at site 5 compared to sites 4 and 6 ($p < 0.05$, Figure 2C), which was due to the lower yields (Figure 1A). In addition, even though we applied 40 kg N ha$^{-1}$ less in 2020 than in 2019 according to recommendation of the French COMIFER method, the NUE in 2020 was globally lower by 27.7%. However, there were contrasting responses depending on the site: in 2019, NUE was high at site 1, intermediate at site 3 and low at site 2, while in 2020 it was higher at site 5 than at sites 4 and 6 (Figure 2D).

Furthermore, following VNT4 application we observed no consistent change in N content in the grain and straw (Table 2). Interestingly, VNT4 significantly improved total grain N by 9.8 kg N ha$^{-1}$ (+6.1%, Table 2 and Figure 2C) and NUE by 2.2 kg kg$^{-1}$ (+3.8%, Table 2 and Figure 2D). Indeed, total grain N was increased by VNT4 irrespective of the site and the year, from 1.3 to 19.2 kg N ha$^{-1}$ (Figure 2C). Considering each year, it seemed that total grain N tended to increase in 2020 (+7.4%, $p = 0.089$, Figure 2C and Table 2). On the other hand, NUE was improved in all situations from 1.4 to 3.8 kg kg$^{-1}$ following the application of VNT4 compared to the control (Figure 2D), and especially in 2019 (+4%, $p = 0.075$, Table 2). Otherwise, application of NBPT increased NUE compared to the control ($p = 0.032$, +2.23 kg kg$^{-1}$, +3.8%) in the same proportions than the VNT4 treatment (Figure S2B).

### 3.2. Experiment 2: Dynamics of Growth, Total N and N Uptake Using the $^{15}$N Labelling Method

#### 3.2.1. Impacts of VNT4 on Growth during the Crop Cycle

In order to better understand the influence of VNT4 on growth and N uptake from soil and fertilizer ($^{15}$N), we conducted a $^{15}$N experiment at site 4 in 2020 with intermediate harvests of aerial parts after each fertilizer application. Compared to the control plots, VNT4 did not significantly change shoot biomass even though the biostimulant slightly improved it at tillering (+6.1%), stem elongation (+9.7%) and the flag leaf extended stage (+10.5%, Table 3). However, between tillering and the flag leaf extended stage, the first two applications of VNT4 significantly increased growth (+10.7%, Figure 3).

**Table 3.** Agronomic and N-related traits at maturity at site 4 in 2020 (Experiment 2 with $^{15}$N-labelling).

|  |  | **Control** | **VNT4** |
|---|---|---|---|
| Aerial biomass (gDW m$^{-2}$) | Tillering | 44.4 ± 4.0 | 47.1 ± 9.7 |
|  | Stem elongation | 348.0 ± 64.8 | 381.7 ± 78.1 |
|  | Flag leaf extended | 867.8 ± 77.0 | 958.6 ± 17.4 |
|  | Maturity | 2041.9 ± 257.3 | 1965.1 ± 159.1 |
| N total in aerial parts (gN m$^{-2}$) | Tillering | 1.57 ± 0.30 | 1.96 ± 0.55 |
|  | Stem elongation | 8.36 ± 1.94 | 9.59 ± 1.65 |
|  | Flag leaf extended | 9.25 ± 0.74 | 10.18 ± 0.91 |
|  | Maturity | 14.27 ± 0.50 | 15.60 ± 0.21 * |
| $^{15}$N total in aerial parts (mgN m$^{-2}$) | Tillering | 6.84 ± 2.45 | 11.16 ± 5.06 |
|  | Stem elongation | 100.00 ± 22.62 | 116.86 ± 19.68 |
|  | Flag leaf extended | 95.84 ± 8.11 | 105.88 ± 7.87 |
|  | Maturity | 126.22 ± 6.08 | 150.74 ± 9.56 * |
| Harvest Index |  | 0.36 ± 0.07 | 0.39 ± 0.05 |
| N Harvest Index |  | 0.71 ± 0.04 | 0.73 ± 0.04 |
| $^{15}$N Harvest Index |  | 0.76 ± 0.04 | 0.80 ± 0.06 |

* denotes significant difference between control and VNT4 according to a pairwise comparison (Fisher's test, $p < 0.05$, n = 4).

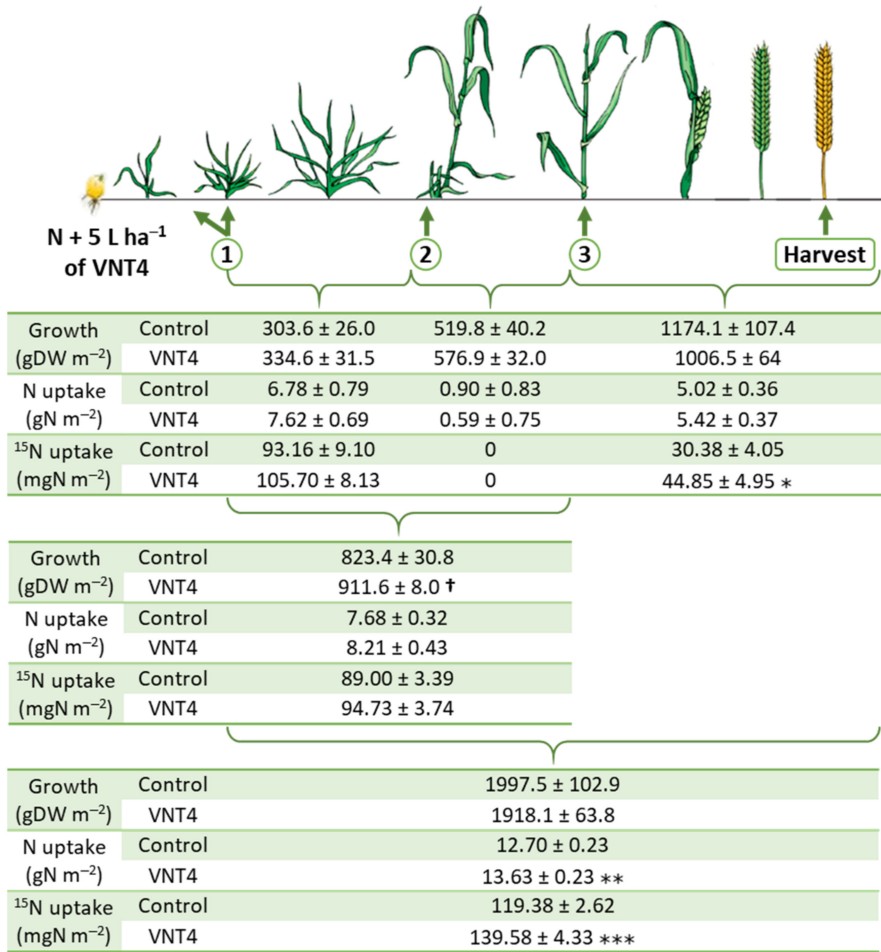

**Figure 3.** Influence of VNT4 on aerial biomass, N and $^{15}$N accumulation in aerial parts between each N and VNT4 application at site 4 in 2020. Plant culture was carried out in France, Normandy. N was provided at the tillering (1), stem elongation (2) and flag leaf stages (3). 5 L ha$^{-1}$ of VNT4 formulation (see Table 1 for more details) were mixed with UAN before each N application, leading to a global application of 15 L ha$^{-1}$. Stars denote significant differences between control and VNT4 according to a pairwise comparison (Fisher's test: *; $p < 0.05$, **; $p < 0.01$, ***; $p < 0.001$, or Kruskal–Wallis test: †; $p < 0.05$, n = 16).

### 3.2.2. Effects of VNT4 on Total N and $^{15}$N Uptake during the Crop Cycle

In addition, the total N in aerial parts was significantly enhanced at maturity (+9.3%) due to a better accumulation of N throughout the crop cycle (Table 3). The first supply of N fertilizer (40 kg N ha$^{-1}$) with VNT4 increased the total N by 24.8% at tillering compared to the control, the 40 + 77 kg N ha$^{-1}$ + VNT4 mix by 14.7% at stem elongation and VNT4 applications enhanced total N by 10.0% at flag leaf stage (Table 3). Remarkably, the three applications of VNT4 allowed a significant augmentation of N accumulation in shoots between tillering and maturity (+7.3%, $p < 0.01$, Figure 3). Furthermore, this experiment showed that VNT4 significantly increased $^{15}$N in aerial parts at maturity (+19.4%, Table 3), which highlights that VNT4 improved fertilizer uptake. This effect was observed after each biostimulant application: the total $^{15}$N in shoots was enhanced by 63% at tillering, 16.9% at stem elongation and 10.5% at the flag leaf extended stage (Table 3). Interestingly, the $^{15}$N uptake after the last VNT4 application, and thus during heading, flowering, and seed development, was improved by a significant 47.6% (Figure 3). This biostimulant effect throughout the crop cycle led to the significant strengthening of $^{15}$N uptake between tillering and maturity (+16.9%, $p < 0.001$, Figure 3). Nevertheless, there was no significant change in terms of the HI, NHI and $^{15}$NHI (+8.3%, +2.8% and +5.3%, respectively, Table 3). Therefore, the better growth with VNT4, mainly during the beginning of the cycle,

allowed greater accumulation of N and [15]N during the entire crop cycle, and especially from fertilizer following the last VNT4 application.

### 3.2.3. Impacts of VNT4 on Elemental Contents in Flag Leaves

To determine the effect of VNT4 applications on the mineral status of flag leaves and the relationship of this status to the level of [15]N uptake observed when the flag leaves were extended, 20 flag leaves were separated at the heading, flowering, seed development and maturity stages for elemental analysis. Despite a slight increase at heading (+5.3%) and a decrease during seed development (−7.8%), VNT4 did not significantly induce changes in the N concentration in flag leaves (Figure 4A). The analysis of whole ionome in flag leaves revealed that only S, Mo and Zn contents were modified by VNT4. Indeed, the S concentration was higher with VNT4 between heading and seed development, with the main effect at the heading and flowering stages relative to the control (+31.2% and +22.3%, respectively, $p < 0.05$, Figure 4B). In addition, Zn concentration was also increased by VNT4 application, especially at the flowering stage (+37.6%, $p < 0.01$, Figure 4C). Finally, the Mo concentration in flag leaves was increasingly improved until seed development following VNT4 application (Figure 4D). Indeed, the Mo concentration was strongly enhanced by VNT4 at this stage (+57.3%, $p < 0.01$, Figure 4D). Altogether, these results indicated that VNT4 biostimulant led to a higher accumulation of S, Zn and Mo in flag leaves at different stages of the crop cycle.

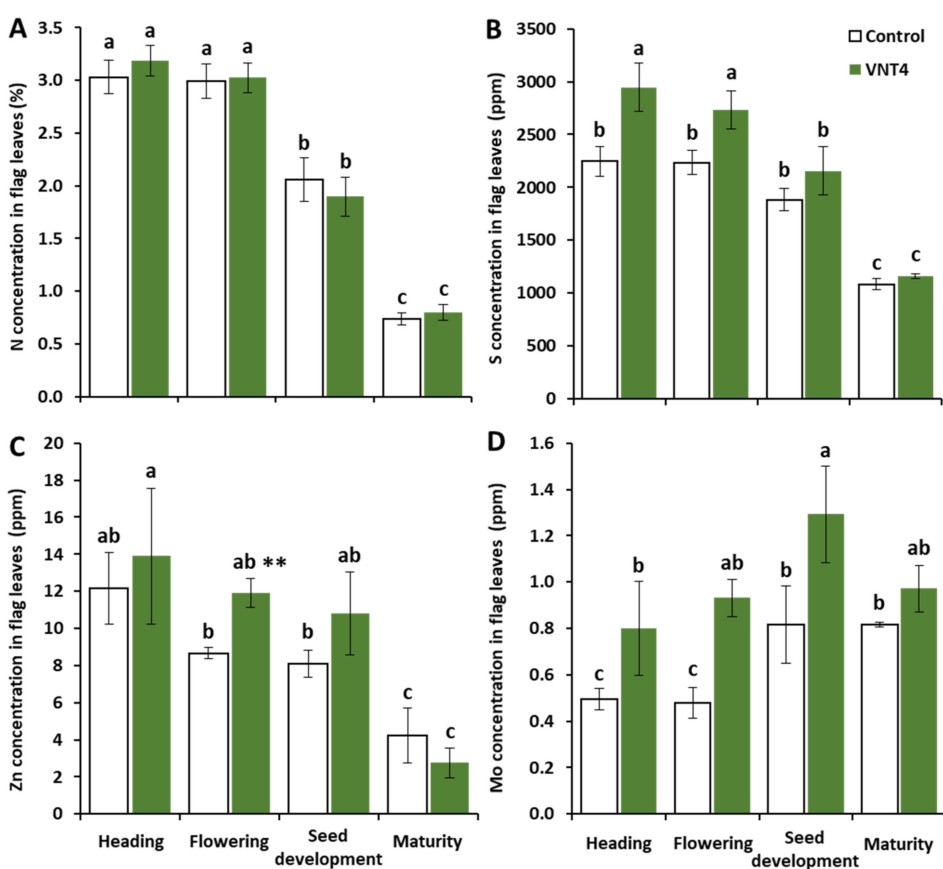

**Figure 4.** Influence of VNT4 on element concentration in flag leaves of winter wheat at site 4 in 2020. (**A**) N concentration, (**B**) S concentration, (**C**) Zn concentration and (**D**) Mo concentration in flag leaves during the crop cycle. N was provided at the tillering, stem elongation and flag leaf stages. 5 L ha$^{-1}$ of VNT4 formulation (see Table 1 for more details) were mixed with UAN before each N application, leading to a global application of 15 L ha$^{-1}$. Bars indicate means ± SE, different letters denote significant differences according to Fisher's test ($p < 0.05$, n = 4) and stars denote significant differences between control and VNT4 according to a pairwise comparison (Fisher's test, **; $p < 0.01$, n = 4).

## 4. Discussion

The aims of the present work were to provide a detailed picture of the effects of VNT4 on grain yield, yield parameters and N-related traits under contrasting field conditions. Clearly, our study highlighted the impacts of this biostimulant on growth, the N dynamics in aerial parts, and the element contents (S, Zn and Mo) of flag leaves and the association of these factors with improved yield, total grain N and NUE (Figure 5).

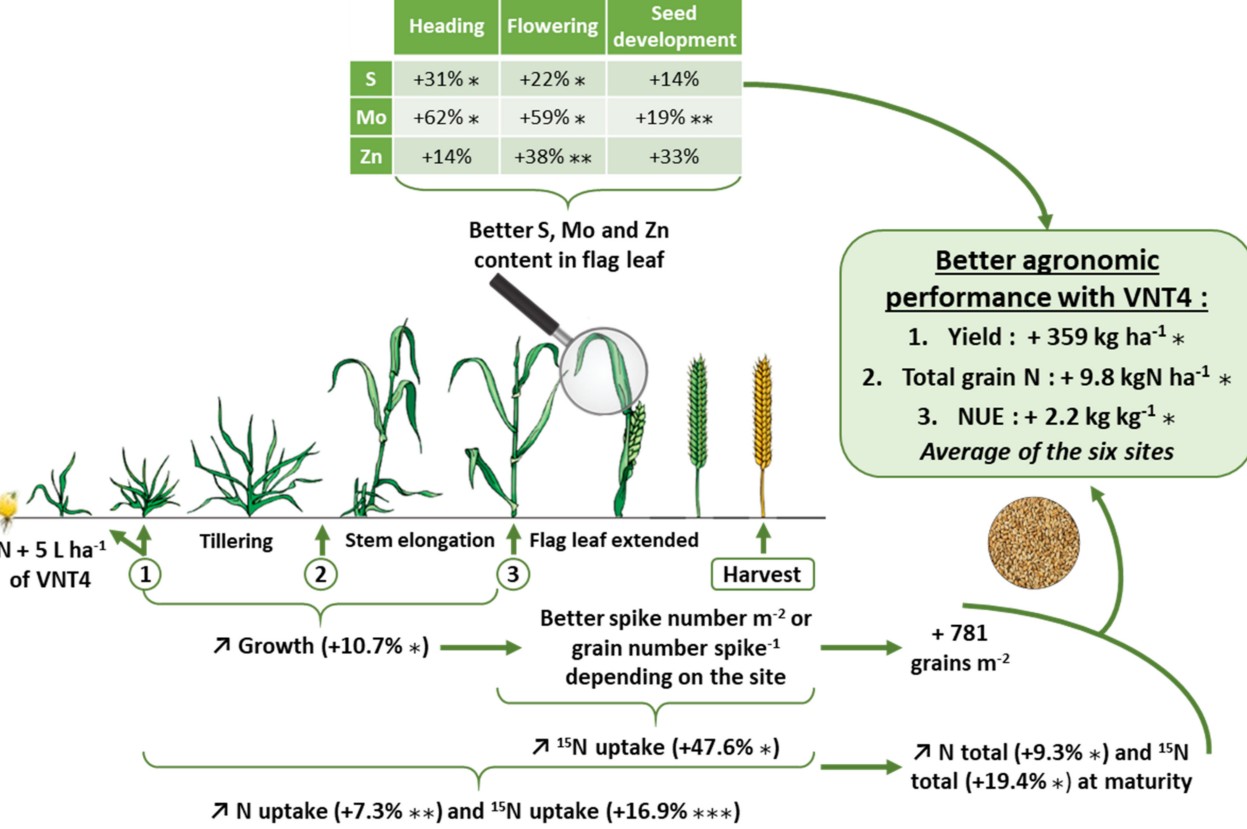

**Figure 5.** Schematic model illustrating the response of wheat to VNT4 supply. Application of VNT4 along N fertilizer improved plant growth during tillering and stem elongation and induced an increase in the spike number per square metre or the grain number per spike, depending on the site. This stimulation enabled production of more grain per square metre. VNT4 also enhanced the accumulation of N and $^{15}$N (especially after the last application) during the crop cycle leading to a higher N and $^{15}$N total in aerial parts at maturity. In addition, application of VNT4 significantly increased the S, Mo and Zn content after the third supply at the flag leaf stage. Taking into account the six sites in 2019 and 2020, the average data revealed better agronomic performance with higher yield, total grain N and NUE when VNT4 was applied to wheat plants. Stars denote significant differences between control and VNT4 (*; $p < 0.05$, **; $p < 0.01$, ***; $p < 0.001$).

### 4.1. VNT4 Improved Yield by Increasing Grain Number per Square Metre and/or TGW Despite Strong Year and Site Effects

In our study, pedoclimatic and agronomic conditions had strong impacts on yield components. Indeed, at each site the previous crop, the wheat cultivar, the sowing date and soil properties were very different (Table S1), inducing variability. Consequently, N fertilizer additions were managed with regard to the field properties and wheat cultivar, which led to greatly contrasted amounts of N supply (from 92 to 210 kg N ha$^{-1}$, Table S2). Weather conditions were also specific to each site: the wheat crops received from 585 mm (at site 5 in 2020) to 883 mm (at site 4 in 2020) of rainfall during the crop cycle and temperatures ranged from −6.1 to 41.7 °C (at site 2 in 2019, Figure S1). The most important rainfall period for wheat is from March to June, when plants need water to grow and take up nutrients such as N. At the same time, in 2020 the wheat crops received

40 kg N ha$^{-1}$ less fertilizer than the recommendation and 17% less precipitation than in 2019 ($-36$ mm during the growing period). These are the two main factors that explain the strong differences in yields and yield components between 2019 and 2020 (Figure 1).

During the two years of experimentation, we also observed a site effect each year: wheat production was lowest at site 3 in 2019 and the lowest production in 2020 was at site 5 (Figure 1A). In 2019, we applied the same amount of N at sites 1 and 3 (152 kg N ha$^{-1}$, Table S2). Therefore, the low harvest at site 3 was probably due to soil type, because superficial clay limestone has less yield potential than clay loams/frank loams (Table S1). The smaller amount of water received at site 3 may also explain this difference in wheat productivity ($-32$ mm compared to site 1). In 2020, the low yields at site 5 were mainly due to the small amount of N applied (92 kg N ha$^{-1}$ while it was more than 170 kg N ha$^{-1}$ at sites 4 and 6, Table S2) and this site received only 152 mm of precipitation during the growing period (whereas it was 208 and 168 mm at sites 1 and 3, respectively).

Despite strong year and site effects on yield components, combining VNT4 with UAN enabled an improvement in grain yield, either by increasing spike number per square metre (sites 1, 3, 4 and 6) or by enhancing grain number per spike (sites 2 and 5), leading to a better grain number per square metre or a higher TGW (sites 3 and 6) (Figure 1). The effects of VNT4 were better in terms of yield in 2019 (+463 kg ha$^{-1}$, +4%) than in 2020 (+255 kg ha$^{-1}$, +3.8%). Indeed, the lowest yield amelioration by VNT4 was observed under the most suboptimal conditions: +129 kg ha$^{-1}$ with only 92 kg N ha$^{-1}$ and 152 mm between March and June at site 2 in 2020. Regarding soil properties, we observed the same results: there were greater increases in yield due to VNT4 in deep silt (+4.3%) than in other soils (+3.7% in clay loams, superficial clay limestone or medium limestone clay). Therefore, even though VNT4 had a positive effect on yield components irrespective of the site and the year, this biostimulant seems to be slightly more efficient under optimal conditions (i.e., higher N and precipitation). For instance, an irrigation in critical times or a larger N fertilization could result in much more yield increases by VNT4.

In order to evaluate the effect of VNT4 relative to NBPT, which is a urease inhibitor that is widely used in Europe, we performed a UAN+NBPT treatment in the six field experiments of this study: NBPT also significantly increased yield compared to the control ($p < 0.05$, +308 kg ha$^{-1}$, +3.4%, Figure S2A) but there was no significant difference between VNT4 and NBPT (Figure S2A, +51 kg ha$^{-1}$ for VNT4). This is in agreement with a meta-analysis on the effect of urease and nitrification inhibitors that concluded that these inhibitors increased crop yields by 7.5% on average [25]. Nevertheless, their effect on yield is variable depending on the molecule used and the targeted crop. For example, several studies showed that NBPT did not improve grain yield in wheat [63–65], even when mixed with NPPT [66]. In contrast, NBPT in combination with *Azospirillum brasilense* inoculation increased wheat yield by 19.6% and 18.8% with application of 100 and 150 kg N ha$^{-1}$, respectively [67]. Moreover, other works have demonstrated the positive impact of urease and nitrification inhibitors on wheat productivity [29,68,69]. Regarding CRFs, a recent meta-analysis indicated that controlled release urea improved grain yield by 5.5% in wheat while the improvements were 8.1 and 7.4% in rice and maize, respectively [35]. On the other hand, biostimulants can enhance wheat yield by up to 17%, depending on weather conditions (optimal conditions or water-deficit stress), such as formulations based on amino acids, *Pseudomonas* strains and plant extracts [70–72]. Application of 5-20 t ha$^{-1}$ biochar also increased wheat grain yield by 2.9$-$19.4% [73]. Finally, our current results are in agreement with our recent findings on Glutacetine®-based formulations, which revealed a significant effect of VNT4 on yield under controlled and field conditions [18]. Based on our previous investigations [18,59], it could be hypothesized that this effect of Glutacetine® formulations on yield is mainly related to better fruiting efficiency, which in turn leads to higher grain number per spike and per square metre. Therefore, the present work confirmed that VNT4 improved wheat productivity to at least the same level as NBPT, as well as CRFs and other biostimulants, whereas it seems to be more difficult to enhance wheat performance relative to other crops [35].

*4.2. VNT4 Increased Total Grain N and NUE Irrespective of the Pedoclimatic Conditions and the N Supply*

As described for yield components, the different weather and fertilization conditions have also impacted N-related traits at each site. In addition to the lower yields, suboptimal conditions in 2020 have resulted in the reduction of N content in grain and straw (Figure 2A,B). Interestingly, with the same N nutrition in 2019 (152 kg N ha$^{-1}$), wheat cultivated at site 3 had a better grain N content than at site 1, whereas the opposite was observed for straw N content (Figure 2A,B). This might be due to the contrasting N remobilization efficiency of the Sacramento (site 1) and Chevignon cultivars (site 3). Nevertheless, total grain N was the same at these two sites and NUE was lower at site 3, suggesting that N uptake efficiency was better with Sacremento (Figure 2C,D). Therefore, Sacremento may absorb more N from the 152 kg N ha$^{-1}$ fertilizer supply to benefit yield (Figure 1A) while Chevignon may be more efficient at remobilizing N from straw towards the grain, which ultimately improved the N content in flour (Figure 2A). This highlights that differences between sites could be due to different cultivars that can contribute towards the crop's NUE [57], but contrasting responses could also be related to site factors (including weather and soil).

Another way to enhance NUE is the use of biostimulants [41] like the Glutacetine®-based VNT4 formulation [18]. Indeed, in accordance with previous findings, the present study confirmed that VNT4 significantly improved total grain N and NUE, irrespective of the site and the year (Table 2). Different kinds of biostimulants, such as marine and fungal extracts, are able to increase total grain N in wheat [44], while others (protein hydrolysate, plant extracts, *Trichoderma*) have enhanced the NUE in various plant species including tomato (*Solanum lycopersicum* L.), dwarf pea (*Pisum sativum* L.) and maize. In the same way, humic acids associated with CRF improved NUE in rice and wheat [33,37], and a biochar-based CRF had a similar effect in oilseed rape [74] and wheat [73]. Interestingly, application of CRF alone has also enhanced the NUE of crops [34,35]. In the present study, the $^{15}$N-labelling experiment carried out at site 4 in 2020 revealed that mixing VNT4 with UAN led to a higher N and $^{15}$N uptake between tillering and maturity (+7.3 and +16.9%, respectively, Figure 5). Remarkably, while there was no $^{15}$N uptake between the stem elongation and flag leaf stages (Figure 3) due to a lack of precipitation (only 12.4 mm during these 20 days), $^{15}$N uptake was improved by 47.6% after the last VNT4 application (Figure 5). This is in accordance with our previous work, which demonstrated that several Glutacetine®-based formulations stimulated post-heading N uptake [18]. Other substances, including marine and fungal extracts and CRFs, increase N uptake [35,44], while urease inhibitors have contrasting results in wheat (no significant impact in four out of five harvests) [27]. However, nitrification inhibitors (DCD and DMPP) tended to improve N uptake in the aboveground parts of the wheat plant [75]. The improvement in N uptake efficiency may lead to higher total N in the plant. Application of VNT4 showed this effect with a significant increase in the total N and total $^{15}$N in aerial parts at maturity (+9.3 and +19.4%, respectively, Figure 5), which is in agreement with our previous study under controlled conditions [18].

Otherwise, from our supplemental data (Figure S2B) it seemed that addition of NBPT increased NUE compared to the control (*p* = 0.032, +2.23 kg kg$^{-1}$, +3.8%) as the VNT4 treatment (2.19 kg kg$^{-1}$, +3.8%). It is well documented that NBPT improves NUE, but it is often a small increase in wheat compared to other crops [25]. For example, Rekowski et al., (2020) reported that urease inhibitor significantly increased NUE by 1.2 kg kg$^{-1}$ (2.3%) and total grain N by 10.5 kg N ha$^{-1}$ (+6.5%), which is in agreement with our results [76]. Therefore, environmentally friendly biostimulants like VNT4 had similar performance to chemical inhibitors and thus could be an alternative for the optimization of grain yield and NUE (Figure 5) in a sustainable agriculture context.

*4.3. The Beneficial Impacts of VNT4 Are also Related to the Enhancement of Plant Growth and Improvements in S, Zn and Mo Contents in Flag Leaves*

Improvement of growth is one the main effects of biostimulants [41]. Indeed, the enhancement of yield and NUE by biostimulants is the result of different processes throughout the crop cycle [42]. In our work, we observed that VNT4 increased growth between tillering and stem elongation (+10.7%, Figure 5), which induced a larger spike number per square metre at site 4 in 2020 (Figure 1B). Several studies have also reported that biostimulants, such as plant extracts and *Bacillus* strains, enhanced wheat growth [48,70]. Moreover, exogenous cytokinin increased leaf photosynthesis, root activity and zeatin content in tiller nodes under low N conditions, which improved tiller bud length [77]. Therefore, in future investigations, it would be interesting to determine how the hormonal balance is affected by VNT4 in regard to its impact on growth and development of reproductive organs.

The strengthening of nutrient uptake efficiency could also be a beneficial outcome of biostimulant application. For example, fluorescent *Pseudomonas* strains present in the rhizosphere increasing P uptake, Si or *Enterobacter* are known to improve Ca and K content, and Fyto-fitness extract has been shown to enhance the Se concentration in wheat plants [71,78–80]. In our field experiment, improvements in the S, Mo and Zn concentrations in flag leaves were observed after the last VNT4 application (Figure 5). This beneficial impact of VNT4 on the S, Zn and Mo status of the flag leaf may have some relationship with the improvement in the physiological function of this organ, which is crucial for sustaining photosynthesis as well as N assimilation and recycling in wheat [81]. Indeed, N accumulation and metabolism is controlled by other elements such as S, Mo and Zn [82–84]. Several reports have provided evidence for strong interactions and especially between S and N metabolism [85–87]. Cysteine, which is the final product of the reductive assimilation of S, and the result of combining sulfide with O-acetyl serine by the cysteine-synthase complex, is an amino acid that represents the convergence of N and S metabolism. Furthermore, it has been reported that S deficiency down-regulates the expression of nitrate reductase and glutamine synthase genes in tobacco (*Nicotiana tabacum* L.) [88]. In fact, when wheat plants suffer from S deficiency, photosynthesis in the flag leaf is limited [89], and grain yield components [90] as well as grain quality [91] are severely impacted. On the other hand, increases in the S content in wheat plants promotes greater shoot biomass and NUE [92]. This is why S is often coupled with N fertilizers to ensure agronomic performance in a number of crop species [16,84,93]. Therefore, a high S concentration in flag leaves may contribute towards maintaining photosynthetic activity and in turn improving NUE and yield. Thus, the significant improvement in the S concentration in flag leaves by VNT4 at the heading and flowering stages balanced the N/S ratio and induced better NUE (Figure 5). This biostimulant effect on S concentration in leaves has also been reported in oilseed rape and in tomato [94,95].

In winter wheat cultivated under different N sources, it was recently reported that Mo strengthened the photosynthetic apparatus and induced the N assimilatory pathway, increasing $NO_3^-$ assimilation and therefore ammonium, amino acid, protein and total N contents [14,82]. The increase in Mo concentration in flag leaves during seed development by VNT4 may help wheat plants to maintain photosynthesis and assimilate more N (Figure 5). This is in accordance with the recent study of Su et al., (2019), which showed that the application of the fungal endophyte *Phomopsis liquidambari* improved Mo nutrition, enhancing nitrate reductase activity and increasing growth parameters in peanuts [96].

Finally, Zn is one of the most important micronutrients to influence plant N homeostasis: a sufficient level of Zn in wheat improves the expression of nitrate transporter genes, amino acid concentrations and glutamine synthetase activity under low N and high N conditions, thus facilitating N accumulation [83]. Besides, an adequate N supply also enhances Zn uptake and root to shoot translocation in winter wheat [15], highlighting the synergistic effect between these elements. Application of VNT4 increased Zn concentration at the flowering stage (Figure 5) while the addition of a microbially derived organic biostimulant resulted in the enhancement of Zn concentrations in the petiole vascular system in

sunflower [97]. In spring wheat, *Bacillus* strains are also able to improve Zn uptake under greenhouse conditions [48]. Therefore, the improvement in S, Mo and Zn concentrations in flag leaves may help to maintain the physiological functions of leaves and sustain growth and N uptake after the last VNT4 application, which led to enhanced total N in the shoots and higher total grain N and NUE (Figure 5).

## 5. Conclusions

Our work analysed the effects of VNT4, a derived formulation of Glutacetine® biostimulant, mixed with UAN on the growth, N-related traits and agronomic performance of winter wheat under six field conditions over two years in Normandy (France), including a site with $^{15}$N labelling. This study has established that VNT4 mixed with UAN under contrasting field conditions leads to a better yield in winter wheat by increasing spike number per square metre, grain number per spike (therefore, grain number per square metre) and/or TGW. Total grain N and NUE are also significantly improved by VNT4, irrespective of the pedoclimatic conditions. These improvements are due to better growth between tillering and the flag leaf extended stages, and to higher S, Mo and Zn content in flag leaves at the heading, flowering and seed development stages. The amelioration of these physiological parameters by VNT4 permitted the uptake of more N fertilizer (+47.6%, Figure 5) in the period from the flag leaf extended stage to maturity. Thus, application of VNT4 increased the N uptake between tillering and maturity, especially N derived from fertilizer, and thus enhanced the total N in shoots at maturity. Finally, all these improvements during the crop cycle induced better agronomic performance: +359 kg ha$^{-1}$ in yield, +9.8 kg N ha$^{-1}$ in total grain N and +2.2 kg kg$^{-1}$ in NUE (Figure 5). VNT4 is also more efficient under optimal conditions in wheat, a crop whose productivity is difficult to improve. Therefore, environmentally friendly biostimulants like VNT4 have similar results to chemical inhibitors when mixed with N fertilizer and thus could be an alternative to improve yield, total grain N and NUE in winter wheat in a sustainable agriculture context.

**Supplementary Materials:** The following are available online at https://www.mdpi.com/article/10.3390/agronomy11051013/s1, Figure S1: Weather conditions for each site during the growing period, Figure S2: Influence of a urease inhibitor (NBPT) and VNT4 on yield and NUE in winter wheat under different site and year conditions, Table S1: Field sites and soil properties of the six experiments, Table S2: Management of N fertilizer inputs for the six field experiments.

**Author Contributions:** Conceptualization, V.M., P.G. and J.-C.A.; methodology, V.M. and J.-C.A.; software, V.M.; validation, V.M. and J.-C.A.; formal analysis, V.M.; investigation, V.M., R.C. and J.-C.A.; resources, V.M., P.G. and J.-C.A.; data curation, V.M. and R.C.; writing-original draft preparation, V.M. and J.-C.A.; writing-review and editing, V.M. and J.-C.A.; visualization, V.M.; supervision, P.G. and J.-C.A.; project administration, V.M., P.G. and J.-C.A.; funding acquisition, V.M., P.G. and J.-C.A. All authors have read and agreed to the published version of the manuscript.

**Funding:** This research and the APC were funded by ANRT (Association Nationale de la Recherche et de la Technologie), which has supported this work as part of a PhD thesis (CIFRE 2017/1239), and by the Région Normandie and FEADER, which supported the FIELD-Prot project (EIP-AGRI).

**Institutional Review Board Statement:** Not applicable.

**Informed Consent Statement:** Not applicable.

**Data Availability Statement:** The data presented in this study are available on request from the corresponding author.

**Acknowledgments:** The authors deeply thank Régis Vecrin, Pierre-Vincent Protin and Frédéric Cardon for their involvement in this work and their valuable advice. The authors acknowledge Laetitia Mabire, Marine Louargant, Thibaut Chouanneau, Romain Laureau, Thomas Simon and Olivier Rose for helping with plant culture, measurements and harvesting. We are most grateful to the PLATIN' (Plateau d'Isotopie de Normandie) core facility for all the elemental analyses done by IRMS and the HR ICP-MS used in this study. We thank Laurence Cantrill for improving and correcting the English in the manuscript.

**Conflicts of Interest:** Authors Victor Maignan and Patrick Géliot were employed by the company Via Végétale. The remaining authors declare that the research was conducted in the absence of any commercial or financial relationships that could be construed as a potential conflict of interest. The funders had no role in the design of the study; in the collection, analyses, or interpretation of data; in the writing of the manuscript, or in the decision to publish the results. The authors declare no conflict of interest.

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
