# Peer review of "VNT4, a Derived Formulation of GlutacetineĀ® Biostimulant, Improved Yield and N-Related Traits of Bread Wheat When Mixed with Urea-Ammonium-Nitrate Solution"

_agronomy, doi:10.3390/agronomy11051013_

Round 1

Reviewer 1 Report

Review

of article: Agronomy-214702

 VNT4, a derived formulation of Glutacetine® biostimulant, im-2 proved yield and N-related traits of bread wheat when mixed 3 with urea-ammonium-nitrate solution

The publication is very interesting, due to the possibility of improving the influence of nitrogen by adding biostimulant to it.

The publication is not objectionable in substance. All chapters are correctly written.

In terms of editing, the work is written at a very high level. My only point concerns Fig.4.

- Figure 4. Please add letters to the graph, which are the difference in post-hoc analysis.

Author Response

Figure 4. Please add letters to the graph, which are the difference in post-hoc analysis.

We add letters to the graph (Figure 4 in revised version) to show the difference in post-hoc analysis.

Reviewer 2 Report

This is a thorough and very well written study. I only have very few minor suggestions and corrections to suggest.

One main point I would suggest is to emphasize it bit more the fact that even though the bio stimulant significantly increased yield, the effect was quite small, especially when compared tool site or year effects. Understanding what induced the differences between years and sites (i.e. water, soil, nitrogen fertilization) could lead to much better improvements. For instance, an irrigation in critical times and critical years, or larger N fertilizations, could result in much more glamorous yield increases.

Was the nitrogen content of the biostimulant accounted for in the various calculations?  This amount was probably negligible given the less than 1% nitrogen in content, which, even with applications of 15 liters per hectare, amounts to very little nitrogen. Therefore, it is totally OK if this amount was neglected in the calculations. But for the sake of clarity, that text could include a small statement clarifying that indeed the nitrogen supplied with the biostimulant was negligible and cannot explain the results.

In the material and methods, it is said that in 2020 fertilization was reduced by 40 kilograms of nitrogen. At first, this appears to men that doses in 2020 were like in 2019, minus 40 kg, but this is not the case. This probably means that 40 kilograms were subtracted not from previous year doses, but from the optimal amount calculated for each site in 2020, with the methodologies described. This is not immediately clear, and it might be worth explaining better.

Describe a bit the “new N requirement indicators 153 (“bq” coefficient)”. Even though a citation is reported (17), some basic understanding should be provided as done for the other method (Méthode du Bilan).

For the N 15 experiment, plots were only one square meter? Then 0.25 square meters were harvested at each sampling? Does it mean that, in four sampling times, the whole square meter was harvested? Didn't this create strong border effects? This part of M&M should be detailed a bit more clearly.

Why were only S, Mo and Zn Evaluated, and not other elements contained in the bio stimulant? Could some of these other elements also have a positive role?

About this statement:

As wheat was under optimal conditions in 2019 and under suboptimal conditions in 2020 (40 kg N ha−1 less than the recommendation), and total N supply was different at each site, significant variations (in yield) between sites were expected.

The fact that total nitrogen supply was different, should not be an explanation for different yields, if the nitrogen supply was purposely different, because it was calculated with appropriate methods to estimate the actual need. Therefore, only the subtracting of 40 kilograms from these calculations should result in lower yields, not different amounts across sites within each year. If, instead, yield was proportional to the nitrogen supply, does this mean that the methods used to calculate the actual nitrogen requirement are not adequate? And that greater yields can be easily achieved by increasing fertilization? And that these increases can be much larger than those achievable by using the bio stimulant? See point above about better improvements with irrigation and fertilization… This point applies also two line 422 where, again, low nitrogen application is reported to cause lower yields.

In line 356 an increase of 6.3% is reported. This should be corrected to 63% .

Line 562: report the full name of P. liquidambari.

Author Response

One main point I would suggest is to emphasize it bit more the fact that even though the bio stimulant significantly increased yield, the effect was quite small, especially when compared tool site or year effects. Understanding what induced the differences between years and sites (i.e. water, soil, nitrogen fertilization) could lead to much better improvements. For instance, an irrigation in critical times and critical years, or larger N fertilizations, could result in much more glamorous yield increases.

In agreement with this comment, we have emphasized this point in the discussion section: “For instance, an irrigation in critical times or a larger N fertilization could result in much more yield increases by VNT4.” (L. 462-464)

Was the nitrogen content of the biostimulant accounted for in the various calculations?  This amount was probably negligible given the less than 1% nitrogen in content, which, even with applications of 15 liters per hectare, amounts to very little nitrogen. Therefore, it is totally OK if this amount was neglected in the calculations. But for the sake of clarity, that text could include a small statement clarifying that indeed the nitrogen supplied with the biostimulant was negligible and cannot explain the results.

We added this sentence to clarify this point: “Because the N content given by the biostimulant was negligible, it was neglected for the calculation of N fertilization.” (L. 151-153)

In the material and methods, it is said that in 2020 fertilization was reduced by 40 kilograms of nitrogen. At first, this appears to men that doses in 2020 were like in 2019, minus 40 kg, but this is not the case. This probably means that 40 kilograms were subtracted not from previous year doses, but from the optimal amount calculated for each site in 2020, with the methodologies described. This is not immediately clear, and it might be worth explaining better.

As suggested, we have modified this sentence to clarify this point: “In 2020, we reduced the dose by 40 kg N ha−1 from the optimal amount calculated to create suboptimal N conditions” (L. 155-156)

Describe a bit the “new N requirement indicators 153 (“bq” coefficient)”. Even though a citation is reported (17), some basic understanding should be provided as done for the other method (Méthode du Bilan).

We detailed this sentence: “For the last N application, we followed the new N requirement indicators (“bq” coefficient) indicating the amount of N to apply after stem elongation for growing each bread wheat cultivar with the dual objective of optimum yield and grain protein content in line with market requirements [17].” (L. 159-160)

For the N 15 experiment, plots were only one square meter? Then 0.25 square meters were harvested at each sampling? Does it mean that, in four sampling times, the whole square meter was harvested? Didn't this create strong border effects? This part of M&M should be detailed a bit more clearly.

We harvested one square meter which was included in bigger plot (1.5 m x 1 m) to avoid border effects. We detailed this point in the text: “Plot size was 24 m² (12 m x 2 m) and a subplot (1.5 m x 1 m) inside this plot was treated with 15N solution.” (L. 174-175)

Why were only S, Mo and Zn Evaluated, and not other elements contained in the bio stimulant? Could some of these other elements also have a positive role?

We have evaluated the whole ionome of flag leaves but the biostimulant have significant effects only on S, Mo and Zn. We have mentioned this information in the revised manuscript in the Materials and Methods: “The whole ionome in wheat flag leaves was quantified by inductively high-resolution coupled plasma mass spectrometry….” (L. 204) ; and in the Results : “The analysis of whole ionome in flag leaves revealed that only S, Mo and Zn contents were modified by VNT4. Indeed, the S concentration was higher with VNT4 between heading and seed development,…” (L.399-401)

About this statement:

As wheat was under optimal conditions in 2019 and under suboptimal conditions in 2020 (40 kg N ha−1 less than the recommendation), and total N supply was different at each site, significant variations (in yield) between sites were expected.

The fact that total nitrogen supply was different, should not be an explanation for different yields, if the nitrogen supply was purposely different, because it was calculated with appropriate methods to estimate the actual need. Therefore, only the subtracting of 40 kilograms from these calculations should result in lower yields, not different amounts across sites within each year. If, instead, yield was proportional to the nitrogen supply, does this mean that the methods used to calculate the actual nitrogen requirement are not adequate? And that greater yields can be easily achieved by increasing fertilization? And that these increases can be much larger than those achievable by using the bio stimulant? See point above about better improvements with irrigation and fertilization… This point applies also two line 422 where, again, low nitrogen application is reported to cause lower yields.

In agreement with this comment, we modified this sentence as follows: “As wheat was under optimal conditions in 2019 and under suboptimal conditions in 2020 (40 kg N ha−1 less than the recommendation), significant variations in yields between years were expected.” (L. 248-250)

In line 356 an increase of 6.3% is reported. This should be corrected to 63%.

Your right, we corrected this point (L. 374).

Line 562: report the full name of P. liquidambari.

We reported the full name: “Phomopsis liquidambari” (L. 592)

Reviewer 3 Report

Dear Authors,

I have studied your article and am of the opinion that it may be published in Agronomy journal. However, it will be necessary to make some changes and edit the article. Because in its current form it contains some inaccuracies and potential mistakes. The presented topic is very current and important, but for admission to a scientific journal, the manuscript should be improved.

General overview

  • The presented work deals with the current topic.
  • This article contains five figures, three graphs and two diagrams/schemas. The schemes can be considered original and increasing the dissert ability of the whole work.
  • The aim/goal of the work is clearly defined. The weak point of the work can be considered a large amount of data. Another potential problem is the description of the measured results, which is not always understandable.
  • This article is very comprehensive because it describes two field experiments.
  • The conclusions should be interpreted with caution, as this is only a two-year field experiment.

I recommend minor revision, but I would like to have the opportunity to comment on the changes made to the article.

Abstract

In the abstract you state (Line 16): … VNT4 significantly improved grain yield (+359 kg ha−1) … This is interesting information, but the results on page 7 show no difference between the control variant and the VNT4 variant. It is the relevance of the overall averages for two years? The result of the total Anova? Please add to the text. Above all why didn't you prepare a graphical abstract?

Introduction

This part is brief and clear - general aspects of the topic are described. This chapter contains quite a lot of information about NUEs and bio stimulants, while relatively little about reactive nitrogen. Reactive nitrogen represents a major threat, for example, in terms of contamination of surface and groundwater sources, see (for example):

  • Sutton et al., The European nitrogen assessment. Cambridge University Press, 2011.
  • Plosek et al., Leaching of mineral nitrogen in the soil influenced by addition of compost and N-mineral fertilizer. Acta Agriculturae Scandinavica, Section B — Soil & Plant Science, 2017.
  • Kichman, H. and Bergstrom L. Do organic farming practices reduce nitrate leaching? Communn Soil Sci Plant Anal, 2001.

Methods

2.1 Field Experiment Design under Contrasting Conditions (Experiment 1)

I have problem with the information about experimental design. I believe that tables describing individual sites should be part of the methodology (not as an attachment). Furthermore, Table S1 lacks information that each experimental area was divided to achieve three replicates. I would also recommend inserting information on climatic conditions.

2.2. 15N Experiment under Field Conditions (Experiment 2, Site 4)

For beater clarity, it would be appropriate to add a map or scheme of the experiment with labelled nitrogen to the methodology.

 2.3. 2.6

Other chapters sufficiently describe the methods used for the analysis of plant biomass and statistical evaluation of the measured data.

Results

3.1. Experiment 1: Study of the Impact of VNT4 on Grain Yield, Yield Components and N- related Traits under Six Contrasting Field Trials

This chapter is divided into two subchapters. The first describes the influence of selected factors (site year and application of VNT4) on grain yield. The second subchapter describes the effect the above factors on N-Related traits.

Subchapter 3.1.1

  • Lines 232 – 251 describe the measured values and their course. I don't have comments here. But since line 265 to line 278 I found some inaccuracies. You state: … global ANOVA revealed a significant effect of VNT4 on yield (p = 0.036, Table 2), which increased by 463 kg ha−1 in 2019 (+4.1%) and by 255 kg ha−1 in 2020 (+3.8%) …   
  • In my opinion, it is not possible to draw such a conclusion from the values given in Table 2. Furthermore, the values for individual sites and years (2019 and 2020) do not show a significant effect of VNT4 application on yield. I recommend to check the statistical data analysis and possibly explain how it is possible in the absence of significant differences in individual years to measure overall significant difference.

Subchapter 3.1.2

  • The description of measured values from individual sites is probably correct, but the overall evaluation of the data is again wrong in my opinion, if Table 2 in the column Treatment and SxT contains values of “p”.
  • Line 313: … the NUE in 2020 was globally lower by 27.7 % (Figure 2D)… Carefully, the Figure 2D does not include the total average values for all sites and the whole year (2019 or 2020).

3.2. Experiment 2: Dynamics of Growth, Total N and N Uptake Using the 15N Labelling Method

This chapter is again divided into several subchapters. The first deals with the influence of VNT4 on the growth of the model crop, the second describes the influence of VNT4 application on N uptake by the plant and the third describes how the application of VNT4 affected the content of microelements in the model plant.

Subchapter 3.2.1

  • Lines 335 – 341: Again, the authors state that the application of the VNT4 had a demonstrable effect on biomass production. However, this claim is not substantiated by the results, Figure 3 does not contain any significant differences in the case of biomass. I recommend checking the data.

Discussion

Line 393: Figure 5. Schematic model illustrating the response of wheat to VNT4 supply.

  • The scheme contains information on a significant increase in grain yield, but it is not clear whether these are data for the entire duration of the field experiment or individual years, individual sites? This significance, however, is not apparent from the measured data.

Subchapter 4.1, lines 403 – 438: This is a description of the data, not a discussion.

  • Here, the discussion should be supplemented by other literary sources and their comparison with the measured data.

Conclusion

This chapter summarizes the measured results and contains a clear conclusion of the whole experiment. However, I recommend the authors to check the measured values. For example:

Line 584: Total grain N and NUE are also significantly improved by VNT4.

  • This is, in my opinion, only partially true, because the graph in Figure 2 shows only partial differences in Total grain N and NUE and Table 2 does not show the significance of the above differences.

Author Response

Abstract

In the abstract you state (Line 16): … VNT4 significantly improved grain yield (+359 kg ha−1) … This is interesting information, but the results on page 7 show no difference between the control variant and the VNT4 variant. It is the relevance of the overall averages for two years? The result of the total Anova? Please add to the text. Above all why didn't you prepare a graphical abstract?

This is the overall averages for two years and the result of the global ANOVA. This point is clarified as follows: “Taking into account all the sites, we report that VNT4 significantly improved grain yield (+359 kg ha−1), total grain N and NUE” L. 16-17. We could prepare a graphical abstract based on Figure 5 if it’s necessary, but we think this figure highlights the mains points in detail.

Introduction

This part is brief and clear - general aspects of the topic are described. This chapter contains quite a lot of information about NUEs and bio stimulants, while relatively little about reactive nitrogen. Reactive nitrogen represents a major threat, for example, in terms of contamination of surface and groundwater sources, see (for example):

  • Sutton et al., The European nitrogen assessment. Cambridge University Press, 2011.
  • Plosek et al., Leaching of mineral nitrogen in the soil influenced by addition of compost and N-mineral fertilizer. Acta Agriculturae Scandinavica, Section B — Soil & Plant Science, 2017.
  • Kichman, H. and Bergstrom L. Do organic farming practices reduce nitrate leaching? Communn Soil Sci Plant Anal, 2001.

Our work is based on biostimulant effect for improving NUE, this is why we emphasized on these topics in the introduction. We did not precisely measure the effect of this biostimulant on leaching of mineral N. Therefore, we think that our description regarding the nitrous oxide is sufficient (see L. 32-37 in the Introduction).

Methods

2.1 Field Experiment Design under Contrasting Conditions (Experiment 1)

I have problem with the information about experimental design. I believe that tables describing individual sites should be part of the methodology (not as an attachment). Furthermore, Table S1 lacks information that each experimental area was divided to achieve three replicates. I would also recommend inserting information on climatic conditions.

We think that tables describing individual sites are essential but as it is not a result, we could not insert these tables in the article (it will be to long). The experimental design is described in materials in methods including the main blocks (see section 2.1. and Table S1) and climatic conditions are described in details in Figure S1.

2.2. 15N Experiment under Field Conditions (Experiment 2, Site 4)

For beater clarity, it would be appropriate to add a map or scheme of the experiment with labelled nitrogen to the methodology.

In agreement with this comment, we have given more details about the 15N labelling method in the revised version of Materials and Methods: see L. 173 to 175 (“Plot size was 24 m² (12 m x 2 m) and a microplot (1.5 m x 1 m) inside this plot was treated with 15N solution”). Based on these details, we think that this methodology is not very complicated and that it is not necessary to add another scheme.

Subchapter 3.1.1

  • Lines 232 – 251 describe the measured values and their course. I don't have comments here. But since line 265 to line 278 I found some inaccuracies. You state: … global ANOVA revealed a significant effect of VNT4 on yield (p = 0.036, Table 2), which increased by 463 kg ha−1 in 2019 (+4.1%) and by 255 kg ha−1 in 2020 (+3.8%) …   
  • In my opinion, it is not possible to draw such a conclusion from the values given in Table 2. Furthermore, the values for individual sites and years (2019 and 2020) do not show a significant effect of VNT4 application on yield. I recommend to check the statistical data analysis and possibly explain how it is possible in the absence of significant differences in individual years to measure overall significant difference.

As recommended by the reviewer, we have carefully checked this statistical data and we confirmed that there is a significant global effect even if the values for individual sites and years (2019 and 2020) did not show a significant effect of VNT4 (p < 0.05). We could observe a strong tendency in 2019 (p = 0.073) and VNT4 slightly increased yield in all sites. Then, taking into account all the data from the two years of experiment, the global analysis indicates a significant effect.

Subchapter 3.1.2

  • The description of measured values from individual sites is probably correct, but the overall evaluation of the data is again wrong in my opinion, if Table 2 in the column Treatment and SxT contains values of “p”.
  • Line 313: … the NUE in 2020 was globally lower by 27.7 % (Figure 2D)… Carefully, the Figure 2D does not include the total average values for all sites and the whole year (2019 or 2020).

Thanks for this valuable comment. Indeed, if there is a value of “p” in Table 2, we could only conclude with a tendency but not to a significant effect. As the Figure 2D does not include the total average values for all sites and the whole year (2019 or 2020), we removed this precision in the revised manuscript (L. 330).

Subchapter 3.2.1

  • Lines 335 – 341: Again, the authors state that the application of the VNT4 had a demonstrable effect on biomass production. However, this claim is not substantiated by the results, Figure 3 does not contain any significant differences in the case of biomass. I recommend checking the data.

We have checked the data and confirmed that VNT4 significantly increased growth (+10.7%, Figure 3) between tillering and the flag leaf extended stage according to a Kruskal-Wallis test. This is the only significant result as indicated in Figure 3.

Line 393: Figure 5. Schematic model illustrating the response of wheat to VNT4 supply.

  • The scheme contains information on a significant increase in grain yield, but it is not clear whether these are data for the entire duration of the field experiment or individual years, individual sites? This significance, however, is not apparent from the measured data.

This is a global mean. We detailed this point in Figure 5 as “Average of the six sites” (See revised version fo figure 5) and in the caption of figure 5 : “Tacking into account the six sites in 2019 and 2020, the average data revealed better agronomic performance with higher yield, total grain N and NUE when VNT4 was applied to wheat plants.” (L. 423-424).

Subchapter 4.1, lines 403 – 438: This is a description of the data, not a discussion.

  • Here, the discussion should be supplemented by other literary sources and their comparison with the measured data.

We believe that the comparison of the differences “year by year” and “site by site” is very interesting in order to understand and discuss what parameters (mainly N fertilization, precipitation and soil type) led to these significant differences. As suggested by the reviewer, we have mentioned some relevant references in this part of the discussion (L. 436, 439 and 445).

Conclusion

This chapter summarizes the measured results and contains a clear conclusion of the whole experiment. However, I recommend the authors to check the measured values. For example:

Line 584: Total grain N and NUE are also significantly improved by VNT4.

  • This is, in my opinion, only partially true, because the graph in Figure 2 shows only partial differences in Total grain N and NUE and Table 2 does not show the significance of the above differences.

As for yield, global ANOVA given in Table 2 is obtained from the six sites and showed a significant effect of VNT4 on total grain N and NUE. This is why we concluded that.

Reviewer 4 Report

The manuscript provides a well written description of a multisite multiyear evaluation of the effect of a biostimulant on yield and NUE of winter wheat in norther France. 

There are a number of issues that require attention which I have detailed .  The principal points are

The experimental design could be better described.

The use of two separate statistical analysis, one which includes all sites over two seasons and a second analysis which analyses each year separately should be justified or else just one analysis, preferably the multiyear analysis, should be used.

The discussion incudes results from an experiment not described in the methods section and not mentioned in the results, but which is used to reach a conclusion.

The discussion makes some statements regarding effects of the biostimulant on different soils and under different conditions that cannot be rigorously tested with the data provided, these should be presented as being speculative.

VNT4, a derived formulation of Glutacetine® biostimulant, improved yield and N-related traits of bread wheat when mixed with urea-ammonium-nitrate solution

While the manuscript provides a well written description of a multisite multiyear evaluation of the effect of a biostimulant on yield and NUE of winter wheat in norther France , there are a number of issues that require attention which I have detailed below.

Lines 131-133 this sentence indicates to me that six cultivars were each grown at six sites, when in fact it would appear that only one cultivar was grown at each site.  Please reword to indicate that experiments were carried out at six sites each with a different cultivar.

Lines 136-137 The plots were randomized in three and four main blocks in 2018-2019 and 2019-2020, respectively I did not understand what this sentence indicated, particularly there term ’main blocks’.  The previous sentence has indicated that there were 3 or 4 replicates depending on season.  Perhaps what was intended was that the experimental design was a randomised complete block design?

It also appears that some of the treatments presented here are very similar in nature to treatments presented in one of the authors papers cited in the references . (Plants 2021,10, 456. https://doi.org/10.3390/plants10030456).   Is the data presented here a subset of treatments from a larger experiment?  If yes this should be stated.

Lines 175-177  if4 x 0.25m2 areas were removed from the microplots of 1 m2 indicating that there was no border area present to avoid edge effects how can the authors be confident that edge effects from the untreated area outside the treated area did not impact on the sampled area and also that edge effects resulting from removal of plants at a previous sampling date did not affect subsequent samplings.  Or perhaps I have misunderstood what was done.

Lines 177-179  as this sentence is follows a description of the microplot experiment it seems logical to assume that the flag leaves were taken from the microplots.  Is this correct? If yes then this further adds to the issues outlined in the previous comment.  If the flag leaves were taken from the larger plots then this needs to be stated.

Figure 1 presents data from the S x T interaction and the text largely relies on the global statistical analysis.  However the statistical comparisons (i.e. the letters over the bars) presented in Figure 1 seem to arise from a separate analysis within each season.  This is confusing as often differences that are highlighted in the text are not actually significant if the letters in figure 1 (i.e within a single season analysis) are taken as guidance.  For example line 282-283 indicates that there was a significant difference in TGW at site 6 but Fig 1 indicates that there was no significant difference.  I do not understand why comparisons made using the global analysis were not used when constructing figure 1? 

Line 372  data presented in Fig 4 would be more appropriately described as concentration data rather than content data

Line 422 ‘low yields at site 5 was mainly’  correction ‘ low yields at site 5 were mainly’ or ‘low yield at site 5 was mainly’

Lines 433 to 438  Regarding soil properties, we observed the same results: there were greater increases in yield due to VNT4 in deep silt (+4.3%) than in other soils (+3.7% in clay loams, superficial clay limestone or medium limestone clay). Therefore, even though VNT4 had a positive effect on yield components irrespective of the site and the year, this biostimulant is more efficient under optimal conditions (i.e., higher N and precipitation).

The data presented in Fig 1 indicates that there were no individual site differences in terms of yield so it is unclear how you can conclude that there were differences between soils.  Also you had no variation in N inputs within a site so you cannot conclude that the biostimulant is more efficient under optimal conditions.  This is, at best, speculation and should not be presented as fact

Lines 439-466  There is no mention of this experiment in the methods and no data are presented in the results. Presenting data from new experiments for the first time in the discussion that have not previously mentioned would seem unusual, particularly as these data are used to form a conclusion. Please either add the relevant information to the methods and results section or remove this section of the discussion.

Lines 480-482  ‘This highlights  the range of strategies present in different wheat cultivars that can contribute towards the  crop’s NUE [57].  The authors correctly speculate that differences between sites may have been due differences in cultivar, but the differences may also have been due to site factors such as weather patterns, soil etc so it cannot be concluded that the results highlight the range of strategies present in different wheat cultivars as no direct comparison can be made between cultivars.  Please reword this sentence to account for this.

Line 486  I am sorry but figure 2c clearly indicates that there is no significant effect of the biostimulant on grain N or NUE at any site.

Author Response

Lines 131-133 this sentence indicates to me that six cultivars were each grown at six sites, when in fact it would appear that only one cultivar was grown at each site.  Please reword to indicate that experiments were carried out at six sites each with a different cultivar.

We rephrased this sentence as: “In 2018 and 2019, experiments were carried out at six sites, each with a different cultivar of winter wheat (Triticum aestivum L., cv. Sacremento, Libravo, Chevignon in 2018; Adoration, Boregar and Extase in 2019) in Normandy, France (see Table S1 for more details), which is classified as having an oceanic temperate climate.” (L. 132-135)

Lines 136-137 The plots were randomized in three and four main blocks in 2018-2019 and 2019-2020, respectively I did not understand what this sentence indicated, particularly there term ’main blocks’.  The previous sentence has indicated that there were 3 or 4 replicates depending on season.  Perhaps what was intended was that the experimental design was a randomised complete block design?

It was the intention, so we replaced this sentence by “The experimental design was a randomized complete block design.” (L. 138)

It also appears that some of the treatments presented here are very similar in nature to treatments presented in one of the authors papers cited in the references. (Plants 2021,10, 456. https://doi.org/10.3390/plants10030456).   Is the data presented here a subset of treatments from a larger experiment?  If yes this should be stated.

Yes, it is. So, according with this comment, we have modified the sentence as follows “The experimental design was a randomized complete block design with 42 and 48 microplots in 2019 and 2020, respectively, and data presented in this work is a subset of treatments.” (L. 138-141)

Lines 175-177 if 4 x 0.25m2 areas were removed from the microplots of 1 m2 indicating that there was no border area present to avoid edge effects how can the authors be confident that edge effects from the untreated area outside the treated area did not impact on the sampled area and also that edge effects resulting from removal of plants at a previous sampling date did not affect subsequent samplings.  Or perhaps I have misunderstood what was done.

In each plot (12 m x 2 m), the harvests (4 x 0.25 m2) were performed in a microplot of 1.5m x 1 m to avoid border effects as well as possible. In order to clarify this point, details are given in the revised manuscript: “Plot size was 24 m² (12 m x 2 m) and a microplot (1.5 m x 1 m) inside this plot was treated with 15N solution”.” (L. 173-175)

Lines 177-179  as this sentence is follows a description of the microplot experiment it seems logical to assume that the flag leaves were taken from the microplots.  Is this correct? If yes then this further adds to the issues outlined in the previous comment.  If the flag leaves were taken from the larger plots then this needs to be stated.

That’s right, the flag leaves were taken from the larger plots. So, we detailed this point: “At the heading, flowering, seed development and maturity stages, 20 flag leaves were separated in the 24 m² plots and at maturity stage, flag leaves, grain and straw were harvested for detailed analyses.” (L. 186-188)

Figure 1 presents data from the S x T interaction and the text largely relies on the global statistical analysis.  However the statistical comparisons (i.e. the letters over the bars) presented in Figure 1 seem to arise from a separate analysis within each season.  This is confusing as often differences that are highlighted in the text are not actually significant if the letters in figure 1 (i.e within a single season analysis) are taken as guidance.  For example line 282-283 indicates that there was a significant difference in TGW at site 6 but Fig 1 indicates that there was no significant difference.  I do not understand why comparisons made using the global analysis were not used when constructing figure 1? 

We decided to separate the statistical treatment for each year because the level of N fertilization is different (optimal in 2019 and suboptimal in 2020). To be clearer, we added stars in the graph when the significant differences was not indicated by the year statistical analysis but by the pairwise comparison, which was the case for TGW at site 6 (See Figure 1 in revised manuscript).

Line 372 data presented in Fig 4 would be more appropriately described as concentration data rather than content data

As recommended, we described Figure 4 as concentration, and we adjusted the text. (L. 399-401, L. 403, 405, 564, 581, 582, 584, 589, 600, 605)

Line 422 ‘low yields at site 5 was mainly’ correction ‘low yields at site 5 were mainly’ or ‘low yield at site 5 was mainly’

We corrected this point as “low yields at site 5 were mainly”. (L. 447)

Lines 433 to 438 Regarding soil properties, we observed the same results: there were greater increases in yield due to VNT4 in deep silt (+4.3%) than in other soils (+3.7% in clay loams, superficial clay limestone or medium limestone clay). Therefore, even though VNT4 had a positive effect on yield components irrespective of the site and the year, this biostimulant seems to be more efficient under optimal conditions (i.e., higher N and precipitation).

The data presented in Fig 1 indicates that there were no individual site differences in terms of yield so it is unclear how you can conclude that there were differences between soils. Also you had no variation in N inputs within a site so you cannot conclude that the biostimulant is more efficient under optimal conditions. This is, at best, speculation and should not be presented as fact.

That’s right, this is speculation. So, we have modified this part as follows: “this biostimulant seems to be slightly more efficient under optimal conditions” (L. 462-463)

Lines 439-466 There is no mention of this experiment in the methods and no data are presented in the results. Presenting data from new experiments for the first time in the discussion that have not previously mentioned would seem unusual, particularly as these data are used to form a conclusion. Please either add the relevant information to the methods and results section or remove this section of the discussion.

As recommended by reviewer, we had a sentence in the Materials and Methods section (L.164-166) about these experiments. The description of these data is also given in the Results section (L. 280-283 and 342-343) and we have adjusted the discussion (L. 467-471 and 537-540).

Lines 480-482 ‘This highlights the range of strategies present in different wheat cultivars that can contribute towards the crop’s NUE [57].  The authors correctly speculate that differences between sites may have been due differences in cultivar, but the differences may also have been due to site factors such as weather patterns, soil etc so it cannot be concluded that the results highlight the range of strategies present in different wheat cultivars as no direct comparison can be made between cultivars.  Please reword this sentence to account for this.

As suggested, we have modified this sentence as follows: “This highlights that differences between sites could be due to different cultivars that can contribute towards the crop’s NUE [57], but contrasting responses could be also related to site factors (including weather and soil).” (L. 509-511)

Line 486  I am sorry but figure 2c clearly indicates that there is no significant effect of the biostimulant on grain N or NUE at any site.

That’s right, it’s the Global ANOVA (Table 2) which indicated this effect. So, we have replaced Figure 2C by Table 2 in revised text (L. 515).
